# The Relationship between Psychological Suffering, Value of Maternal Cortisol during Third Trimester of Pregnancy and Breastfeeding Initiation

**DOI:** 10.3390/medicina59020339

**Published:** 2023-02-10

**Authors:** Anca Ioana Cristea Răchită, Gabriela Elena Strete, Andreea Sălcudean, Dana Valentina Ghiga, Adina Huțanu, Lorena Mihaela Muntean, Laura Mihaela Suciu, Claudiu Mărginean

**Affiliations:** 1Doctoral School, “George Emil Palade” University of Medicine, Pharmacy, Science and Technology of Târgu Mureș, 540139 Târgu Mureș, Romania; 2Department of Psychiatry, “George Emil Palade” University of Medicine, Pharmacy, Science and Technology of Târgu Mureș, 540136 Târgu Mureș, Romania; 3Department of Ethics and Social Sciences, “George Emil Palade” University of Medicine, Pharmacy, Science and Technology of Târgu Mureș, 540136 Târgu Mureș, Romania; 4Department of Medical Scientific Research Methodology, “George Emil Palade” University of Medicine, Pharmacy, Science and Technology of Târgu Mureș, 540136 Târgu Mureș, Romania; 5Department of Functional and Additional Sciences, “George Emil Palade” University of Medicine, Pharmacy, Science and Technology of Târgu Mureș, 540136 Târgu Mureș, Romania; 6Department of Obstetrics and Ginecology Clinic II, “George Emil Palade” University of Medicine, Pharmacy, Science and Technology of Târgu Mureș, 540136 Târgu Mureș, Romania

**Keywords:** cortisol, LATCH score, breastfeeding initiation, maternal stress, anxiety, depression

## Abstract

*Background and Objectives:* Cortisol, the stress hormone, is an important factor in initiating and maintaining lactation. Maternal suffering during pregnancy is predictive for the initiation and shorter duration of breastfeeding and can also lead to its termination. The aim of this study is to evaluate the relationship between the level of salivary cortisol in the third trimester of pregnancy and the initiation of breastfeeding in the postpartum period in a cohort of young pregnant women who wanted to exclusively breastfeed their newborns during hospitalization. *Materials and Methods:* For the study, full-term pregnant women were recruited between January and May 2022 in the Obstetrics and Gynecology Clinic of the Mureș County Clinical Hospital. Socio-demographic, clinical obstetric and neonatal variables were collected. Breastfeeding efficiency was assessed using the LATCH Breastfeeding Assessment Tool at 24 and 48 h after birth. The mean value of the LATCH score assessed at 24 and 48 h of age was higher among mothers who had a higher mean value of salivary cortisol measured in the third trimester of pregnancy (*p* < 0.05). A multivariate logistic regression model was used to detect risk factors for the success of early breastfeeding initiation. *Results:* A quarter of pregnant women had a salivary cortisol level above normal limits during the third trimester of pregnancy. There is a statistically significant association between maternal smoking, alcohol consumption during pregnancy and the level of anxiety or depression. *Conclusions:* The most important finding of this study was that increased salivary cortisol in the last trimester of pregnancy was not associated with delayed initiation/absence of breastfeeding.

## 1. Introduction

Stress is defined as a process in which the demands of the environment overload the adaptation capacity of an organism, which may lead to both psychological and biological changes [1].

The recommendation of the World Health Organization and the American Academy of Pediatrics to exclusively breastfeed infants for the first 6 months is not met by one in four mothers, and three in five mothers do not meet their own breastfeeding goals [2]. Exclusive breastfeeding until 6 months of age guarantees many health benefits, decreasing the risk of asthma and respiratory diseases [3,4], gastrointestinal infections [5], sudden infant death syndrome [6], ovarian cancer [7], cardiovascular disease and type 2 diabetes for mothers [8,9,10].

More recent studies link maternal psychological distress to unsuccessful breastfeeding [11]. The most studied mechanism to demonstrate the relationship between psychological distress during pregnancy and suboptimal breastfeeding outcome is the activation of the hypothalamic-pituitary-adrenocortical (HPA) axis [12].

Studies on the relationship between stress during pregnancy, initiation and duration of breastfeeding are few, and their methodology and results may be debatable. Consequently, we have identified in the specialized literature studies on the relationship between depression during pregnancy and the initiation of breastfeeding: two of them demonstrated that there is no association between the two variables [13,14] and five other studies demonstrated that depression symptomatology during pregnancy is a predictor for short duration of breastfeeding [15,16,17,18,19]. Another study clearly demonstrates the relationship between depression during pregnancy and the presence of difficulties during breastfeeding [20]. The relationship between postpartum depression and breastfeeding is much debated in studies published in recent years [21,22]. These studies demonstrate that the presence of stress precedes and leads to the early cessation of breastfeeding. Specialized studies show that the diurnal cortisol rhythm is preserved during pregnancy, showing a significant increase in the early morning, followed by a constant decrease throughout the day [23]. Among a cohort of pregnant women included in risk categories, it was found that the maternal cortisol level increased from 390 ± 22 nmol/L on average at the 5th gestation week to 589 ± 15 nmol/L at the 20th gestation week [24]. Further on, obese mothers during early pregnancy are known to have lower cortisol levels in the morning [25]. However, Oaks et al. [26] found that higher maternal cortisol levels in early and mid-pregnancy were related to shorter gestational duration and an increased risk of preterm birth, as measured by at 16, 28 and 36 gestation weeks. Di Pietro et al. monitored salivary cortisol levels in the third trimester weekly and found that it varied by fetal sex, with women carrying male sex fetuses having higher cortisol levels compared to those carrying female sexual fetuses [27]. During gestation, tracking cortisol as a biomarker of stress is particularly difficult because cortisol levels are affected by acute stress, pain and therapeutic interventions. Another study demonstrates that the total cortisol level measured 1 min after delivery was 49.74 µg/dL [28] and reached 81.9 µg/dL among exhausted women due to acute stress associated with childbirth, which shows that cortisol levels are related to physical and emotional stress associated with birth complications. Cortisol is a necessary cofactor for milk production and is involved in the differentiation of mammary gland cells into lactocytes, as well as in milk secretion and lactogenesis [29].

The link between stress, cortisol levels and breastfeeding is what we aim to elucidate in our study. Until now, it is known that maternal stress is frequently encountered in the postpartum period. It is also known that increased levels of cortisol influence the composition of breast milk, respectively, the level of free fatty acids [30]. Because in the early neonatal period, human milk is the only source of free fatty acids, studies addressing the relationship between cortisol levels and successful initiation of natural nutrition in the early postpartum period demonstrate the role that cortisol levels have on the success of breastfeeding initiation [31].

We investigated the hypothesis that maternal stress in the pre-birth period affects the success of breastfeeding initiation by analyzing the relationship between the salivary cortisol level and breastfeeding, assessed by the LATCH score at 24 and 48 h after birth.

Parameters commonly used to assess breastfeeding technique during the initial postpartum period are designated by the acronym LATCH: “L” is for how well the baby latches onto the breast; “A” is for the amount of audible swallowing; “T” is for types of the mother’s nipples; “C” is for the mother’s comfort level; “H” is for the support the mother needs to hold her baby at the breast [32]. The LATCH is a sensitive, reliable and valid instrument that assesses breastfeeding technique based on observations and descriptions of effective breastfeeding [33].

Many studies have demonstrated strong sensitivity, specificity and good predictive value of LATCH score thresholds related to unsuccessful breastfeeding for both vaginal and cesarean delivery [34,35].

This study aimed to assess the relationship between psychological suffering, perceived stress in the last trimester of pregnancy and successful initiation of breastfeeding. The study describes the first relationship between two scientific clinical tools for:(i)Salivary cortisol level;(ii)Quantifying the psychological suffering with Hospital Anxiety and Depression Scale (HADS) screening test;(iii)Successful breastfeeding technique in the initial postpartum period assessed with the LATCH score at 24 h and 48 h after delivery.

### Objectives

The current study focuses on the association between maternal suffering, the level of salivary cortisol (which values faithfully reflect the level of free blood cortisol) determined in the third trimester of pregnancy and the successful initiation of breastfeeding as assessed by the LATCH score in a cohort of hospitalized mothers and newborns in the Obstetrics and Gynecology Clinic of the Mureș County Clinical Hospital. We assumed that a higher level of maternal salivary cortisol will be related to the absence of breastfeeding in the first 48 h after birth.

## 2. Materials and Methods

### 2.1. Study Design

We carried out a prospective observational study, between January and May 2022. The study included 76 pregnant women in the last trimester of pregnancy, recruited for hospitalization in the Obstetrics and Gynecology Clinic of the Mureș County Clinical Hospital. The study took place after obtaining approval from the Institutional Research Ethics Committee (Approval No. 1676, Approval date: 5 January 2022). Of convenience, we chose a sample of 60 pregnant women who expressed their desire for exclusive natural nutrition and accepted inclusion in the study.

### 2.2. Participants and Procedure

A total of 76 mother-child pairs volunteered to participate in this study. According to the inclusion/exclusion criteria, 6 pairs were excluded due to premature birth (before 37 weeks of pregnancy); 4 mothers were excluded due to interruption of breastfeeding (the cause being the mother) within the first 48 h after birth; 6 other pairs were excluded (the newborns who required care in the neonatal intensive care unit (NICU)).

Following the secondary selection according to the inclusion/exclusion criteria, a convenience sample group of 60 pairs of new-born mothers, aged over 16, who expressed their desire for exclusive natural nutrition and accepted inclusion in the study, healthy (without cardiovascular or renal comorbidities) who declare that they choose to breastfeed immediately after birth, from which written consent was obtained. The medical data of patients and newborns were taken from the clinic’s electronic system and then stored in a database and identification. The post hoc analysis indicated this sample size yielded a power of 89.6% with the probability of type I error of 0.05.

Patients hospitalized and included in the study group align the ten steps of the Baby Friendly Hospital initiatives, especially step 3 (inform all pregnant women about the benefits and management of breastfeeding), step 5 (show mothers how to breastfeed and how to breastfeed to maintain lactation) and step 7 (rooming practice—in postnatal care allows mothers and babies to stay together 24 h a day).

Inclusion criteria:Pregnant women who declare that they want to breastfeed immediately after birth;Pregnant women over the age of 16, with secondary education;The signed consent of the mother for the inclusion of the newborn in the study;Only term newborns and without comorbidities are eligible.

Exclusion criteria:Pregnant with preterm delivery;Pregnant women with cardiovascular and renal comorbidities, mental illnesses in the antecedents;Newborns who required maneuvers and intensive therapy treatments at birth.

Because it is not possible to accurately predict the day of birth and because no patient included in the study had a scheduled elective birth, the cortisol level was determined from the saliva sample collected at the time of hospitalization. Hospitalized patients were included only if they were in induced labor, and we only considered cortisol determinations of births that occurred within 0–24 h of saliva collection. Cortisol levels can increase due to hospitalization, pain and the type of birth. Its value represents a conclusive factor for studying the relationship between cortisol levels and breastfeeding success.

To evaluate the success of natural nutrition after birth [36], the LATCH score was evaluated 24 and 48 h after birth, by a single nurse with experience in conducting clinical and research studies, with the aim of minimizing interpretation errors.

The HADS scale has 14 items of which 7 are for anxiety, and 7 are for depression. Each item is evaluated on a Likert scale with 4 anchors, from 0 = never, to 3 = always. The items for anxiety are represented by odd questions (1,3,5,7,9,11,13) and indicate a level of depression rated on a Likert scale. The items for depression are represented by even questions (2,4,6,8,10,12,14) and indicate a level of depression rated on a Likert scale. To rate the answers, the score is calculated for each scale: anxiety, respectively, depression, taking into account the grid, summing up the scores obtained for the items of the scales separately (Table 1).

### 2.3. Measures

The studied variables included for the mother: age, parity, last school completed, level of education and marital status. Age was recorded in whole years at birth. The occupational characteristics extracted from the interview include the work schedule, in order to identify if they also perform night shifts. Maternal medical factors were parity, gestational age and type of delivery. Gestational age was assessed sonographically, and birth weight was recorded on birth records. The neonatal medical factors were Apgar score, need for biological blood sampling and jaundice therapy according to the Bhutani nomograms used to determine the level of risk based on the newborns hours of age [37].

They were also instructed to provide the saliva sample in a collection container (Salivette Cortisol) at least 30 min after physical exertion, consumption of food and liquids, smoking or oral hygiene. Saliva samples were rejected if the time of assessment was not recorded or if the sample was insufficient (below 1 mL). Saliva samples were stored for 3 days at 2–8 °C and then frozen at −20 °C until processing. The samples being “precious and small in volume”, for which the recalculation was quite difficult, we did not have enough volume for the duplicate. Before testing, the samples were thawed, homogenized with the help of a vortex and then centrifuged for 15 min at 3000 RPM to remove any traces of impurities or mucus [38].

The dosage of salivary cortisol was carried out with a kit designed for the quantitative determination of salivary cortisol (DiaMetra, Spello, Italy) using a competitive ELISA (enzyme-linked immunosorbent assay) technique, on the Dynex DSX (Dynex Technologies, Chantilly, VA, USA) automated ELISA immunology analyzer. For testing, the competitive technique was used; in the 96-well plate, salivary cortisol competes with antigenic cortisol (from the reaction kit), labeled with peroxidase. The two analytes compete for a limited number of binding sites in the wells, forming antigen–antibody complexes. The addition of the substrate represented by tetramethylbenzidine (TMB) determines the appearance of a color reaction whose intensity will be inversely proportional to the concentration of the analyte in the sample [38]. The cortisol concentration in each sample is calculated using the reaction curve obtained with seven calibrators. The performance characteristics of the saliva cortisol dosing kit (according to the manufacturer’s statement) were as follows: repeatability (intra-assay precision) ≤ 10%, reproducibility (inter-assay precision) ≤ 8.3%, sensitivity expressed by the lowest detectable concentration of 0.12 ng/mL and a 100% specificity for cortisol, without significant interference with other parameters.

The LATCH score was calculated for each newborn at 24 and 48 h after birth. The tool assigns 0, 1 or 2 for the five parameters [14]. The same grid was used to calculate the LATCH score at 24 and 48 h of age. The LATCH score was assessed during routine day shifts with the neonatologist, the total score ranging from 0 to 10, with 10 representing effective and 0 representing inadequate breastfeeding technique.

The LATCH score, used in this study to measure the success of natural feeding, was published by Jensen, Wallace and Kelsay [39] and was created to provide a systematic method to obtain information about mothers’ natural feeding capabilities. The system grades with 0, 1 and 2 the 5 important items of natural nutrition. A study carried out on a population of 80 mothers of Turkish origin that evaluates the success of natural nutrition after vaginal birth demonstrates a Cronbach alpha coefficient value of 0.96 on day 1 and 0.94 on day two of the LATCH score [39]. The total score ranges from 0 to 10; the higher the score, the greater the chance of breastfeeding success (Table 2). A LATCH score of 0–3 is considered poor, 4–7 moderate, and 8–10 good [40].

Psychological suffering was quantified with the help of the Hospital Anxiety and Depression Scale due to its ease of application and its reliability in detecting anxiety/depression, a scale developed by Zigmond and Snaith in 1983 [41]. Translation in Romanian was first performed by Dr. Radu Teodorescu and published in Sinapse magazine (1996). The Romanian version of the scale shows appropriate reliability and validity. The use of this instrument is strongly recommended to assess anxiety and depression in the chronic pain population, as it does not include items of a somatic nature that can give false positive results. In Romania, the validity of HADS was confirmed in both psychiatric and medical adult patients, and the validation studies showed a high internal consistency of both scales: HADS-A (Cronbach’s α ranging between 0.68 and 0.93) and HADS-D (Cronbach’s α ranging between 0.67 and 0.90) [42]. It was validated on the Romanian psychiatric population by Dr. Maria LADEA, in her PhD thesis [42].

The HADS scale includes 14 items regarding the intensity of anxious and depressive feelings. Each item is rated on a Likert scale with 4 anchors, from 0 = never, to 3 = always. A score from 0 to 7 means the absence of depression/anxiety; a score between 8 and 10 indicates mild signs; above 11 indicates the presence of a case that requires additional attention.

Due to the fact that the HADS scale is a screening test and not a diagnostic test (the diagnosis is made following the evaluation of clinical symptoms in accordance with the DSM-IV-R criteria), the interpretation of the results was as follows: between 0 and 7 points (normal result, we cannot talk about anxiety or depression). Considering the intensity of anxiety and depression symptoms, we divided the group into two: the risk group, with a score greater than or equal to 11, and the non-risk group, with scores less than or equal to 10.

### 2.4. Statistical Analysis

The statistical analysis included elements of descriptive statistics (frequency, percentage, mean, median, standard deviation, correlation coefficient and 95% confidence interval) and elements of inferential statistics. The Shapiro–Wilk test was applied to determine the distribution of the analyzed data series. For the comparison of means, the t-Student test was applied for unpaired data, respectively, the Mann–Whitney test, non-parametric tests for the comparison of medians. Multiple linear regression was applied to evaluate the impact of the independent variables on the dependent variables and the Fisher test to determine the association between the qualitative variables. The significance threshold chosen for the *p* value was 0.05. The dependent and independent variables used in the analysis were the following:Dependent variable: LATCH score;Independent variable: parity, smoking, salivary cortisol level and cesarean delivery.

The statistical analysis was performed using the demo version of the GraphPad Prism utility [43].

## 3. Results

The average age of the pregnant woman was 28.5 (SD ± 6.4) years, min–max (16–40) years, without statistically significant differences between the two groups.

According to the interpretation of the HADS scale, a score above 11 indicates the presence of anxiety or depression. In our group, anxiety scores ranged from 1 to 20 for 25% of participants (n = 15 mothers) with a mean score of SD = 15.25, and depression scores ranged from 1 to 21 with a mean score of SD = 15.36, considering the risk group. In the remaining 75% (n = 45 mothers), mean scores of anxiety SD = 9 and depression SD = 9.1 indicate mild signs or no anxiety/depression SD anxiety = 2.41; depression SD = 2.64 representing the group without risk.

Most of the patients come from the urban environment (71%) and are married (61.6%).

More than half of the mothers included in the study had secondary education (53.33%). We noticed that a large percentage of mothers (46.67%) had higher education, being desirous of effective collaboration. There is a statistically significant association between social status (social status is the position a person occupies within a social group and responds to a hierarchy based on prestige) and the level of anxiety or depression [RR = 0.1818, CI (95%): 0.6616–0.4997, *p* = 0.0003]. Most of the mothers are employed (66.67%), but in the group of mothers who worked night shifts, the average anxiety and depression scores had statistical significance (*p* = 0.0031) Table 3.

To assess the success of natural feeding after birth, the LATCH score was assessed at 24 and 48 h after birth. For the LATCH score as dependent variable, continuous scores were used in the data analysis. We used a binary logistic regression model; a LATCH score less than 8 at 24 and 48 h of age was the dependent variable, and parity, smoking, maternal cortisol level, type of delivery (caesarean) and early skin-to-skin contact were included in the covariates—Table 4.

Adjusted R Square = 4.5% of the variation of the LATCH score/newborn 24 h is explained by the variation of smoking, parity, maternal cortisol values and cesarean delivery.

For smoking mothers, the coefficient is −0.105; in smokers the 24 h LATCH/newborn score will decrease by 0.105.

For parity, the coefficient is −0.213; when parity increases by one unit, the LATCH score/48 h newborn will decrease by 0.213.

For salivary cortisol level, the coefficient is −0.009; when the salivary cortisol level increases by one unit, the LATCH score/newborn 24 h will decrease by 0.009.

For birth by cesarean section, the coefficient is 0.892; for newborns by cesarean section, the LATCH score/newborn 24 h will increase by 0.892.

At 24 h after birth, one of the risk factors associated with failure to initiate natural feeding is birth by cesarean section. The coefficient is 0.892; the LATCH score/newborn 24 h will increase by 0.892, (*p* = 0.207), and early skin-to-skin contact reduces this risk OR 0.238 (*p* = 0.01).

At 48 h after birth, smoking is a risk factor associated with the failure of natural nutrition. The coefficient is 0.616; the LATCH score/newborn at 48 h will increase by 0.616, (*p* = 0.401).

For parity, the coefficient is −0.226; when parity increases by one unit, the LATCH score/newborn 48 h will decrease by 0.226, (*p* = 0.580).

For salivary cortisol level, the coefficient is −0.003; when the salivary cortisol level increases by one unit, the LATCH score/newborn 48 h will decrease by 0.003, (*p* = 0.710).

For birth by cesarean section, the coefficient is 0.438; for newborns by cesarean section, the LATCH score/newborn 48 h will increase by 0.438, (*p* = 0.515).

We analyzed the neonatal characteristics: Apgar score, VG and weight in order to see how neonatal variables can influence success of breastfeeding.

From the neonatal characteristics (Table 5) in the study population, 10% of the newborns were detected with a birth weight lower than the 10th percentile according to sex-specific birth weight charts.

In the group with average scores of birth weight (3139 ± 524.3, 3230) a small difference (*p* = 0.4354) was detected in terms of birth weight for gestational age. For the HADS score, the binary data were used, and based on this data, the subjects were divided in the two groups that were analyzed.

## 4. Discussion

The level of psychological stress during pregnancy, frequently measured as the association of salivary cortisol with perinatal outcome and child health outcome, is an intensively studied subject [44,45,46].

This is the first study to examine the association between third-trimester cortisol levels and breastfeeding in the postpartum period. Although our study focuses on the impact of cortisol on successful initiation of breastfeeding, salivary cortisol level measured in the third trimester was not associated with delayed breastfeeding at 24 h of age, nor at 48 h postpartum, which is the strength of the study. In addition, our study measures a large number of potentially confounding variables of successfully breastfeeding initiation to optimize the association between anxieties, stress and cortisol level with the breastfeeding success in early postpartum period. This result was in line with the findings of Chung et al. [47], observing the effect of psychological factors and nutrient content of human milk at 7–14 days postpartum.

There are other modifiable factors of successful early breastfeeding initiation that are revealed in this study. Early skin contact in relation to maternal smoking can be a risk factor associated with the failure of natural nutrition. In our case, the multivariate logistic model for smoking had a −0.105 coefficient, and the LATCH/newborn score was (*p* = 0.891). However, we found that smoking mothers had shorter breastfeeding durations compared to non-smoking mothers, were significantly younger and had a lower level of education [48].

We detected in the studied group, the socioeconomic predictors for the increased level of cortisol such as the professional activity in night shifts, but the logistic regression analysis did not provide this variable as being a risk factor for the adverse initiation of breastfeeding at 24 or 48 h from birth.

Mean anxiety (mean anxiety = 15.25) and depression (mean depression = 15.36) scores were correlated with elevated cortisol levels (26.37) and yet did not influence breastfeeding initiation as assessed by the LATCH score.

Previous studies demonstrate the robustness of the association between the level of salivary cortisol and the level of stress during pregnancy at different gestational ages [49]. These authors emphasize the importance of the salivary cortisol level, which can be an important predictor of child outcome because it may be the best indicator of HPA axis disturbances caused by maternal stress [50].

According to the results of this study, no relationship was observed between salivary cortisol level and perinatal outcome and birth weight. This observation is supported by the work of Lundholm et al., which suggests that there was no evidence for an association between prenatal maternal psychological measures of stress or salivary cortisol levels with lower birth weight, birth weight for gestational age or gestational age [51]. Furthermore, neither perceived maternal stress during pregnancy, nor maternal cortisol was associated with infant salivary cortisol levels at 3 months of age [52].

Women with Gestational Diabetes have a 4.00 times higher risk of developing Medium Anxiety or Depression, statistically significant [CI (95%): 2.013–7.949, *p* = 0.0118].

For 25% of the mothers the cortisol level [54.96 ± 70.73 (26.37% mean of elevated cortisol)] was higher than the normal range (N = 15, the group with average anxiety and depression scores), and for 75% of the cases the level of cortisol [11.58 ± 5.729 (10.74%) Mean normal cortisol] maternal was in the normal range (N = 45, with mild anxiety and depression scores, respectively, without anxiety and depression). However, an important limitation of this study is that we only have a single measure of cortisol. The gold standard is to take several cortisol samples at specific standardized points in time over the course of 48 h. Previous studies suggest that one salivary cortisol level collected during the morning is adequate for differentiating pregnant women with low or high cortisol level [53]. Performing sequential measurements of salivary cortisol level during third trimester of pregnancy was not feasible in our study; however, the anxiety and depression were measured at the same point in time with salivary cortisol level, which enables us to better capture the relationship between the level of stress with the breastfeeding outcome.

The mean LATCH breastfeeding score was higher in the postpartum period among women with elevated cortisol and was found to be increased from Day 1 to Day 2 in both groups. The results of the study suggest average levels of anxiety and depression, also confirmed by other authors [54], which represent potential risk factors for the subsequent development of a depressive or anxiety disorder. Stress management, through interventions such as counseling, health education and extended care programs (teams consisting of obstetricians, nurses, and specialists in the field of psychological health) are the primary factor in the prevention of postpartum mental disorders, which can improve (reduce) anxiety and depression scores among women and during the prenatal period, from a medical point of view.

For the current study, factors that may influence breastfeeding were included: maternal smoking, type of delivery (vaginal versus cesarean), skin-to-skin contact, parity and maternal education based on perceived importance in clinical practice [55]. At 24 h age assessment, skin-to-skin contact and cesarean delivery were associated with lower risks of breastfeeding initiation (CI for OR: −0.509–2.294), while at 48 h age, parity (CI for OR: −1.063–0.638) and smoking (CI for OR: −0.509–2.294) while at the age of 48 h, parity (CI for OR: −1.063–0.638) and smoking (−1.635–1.424) were −1.635–1.424, and were associated with unsuccessful breastfeeding technique. An important indicator of positive mother–infant bonding is skin-to-skin and/or postnatal care and is associated with increased breastfeeding initiation as well as continued breastfeeding [56,57].

Vaginal birth has been consistently associated with successful initiation and continuation of breastfeeding in several studies [58,59,60]. Cesarean delivery may be associated with unsuccessful initiation of breastfeeding due to disruption of the infant–mother pair and due to decreased oxytocin secretion or maternal stress, which may lead to decreased milk production [61]. Many studies examine the relationship between parity and breastfeeding. While there are authors who show a positive association with multiparity [59,62] others demonstrate a negative effect due to failure to initiate breastfeeding with the first child [63], but in our study, parity was associated only with 48 h breastfeeding outcome.

A limitation of this study was that we did not include information about previous breastfeeding experience, especially initiation of breastfeeding in previous children, in the variables we studied. The ability to assess the impact of previous breastfeeding success on subsequent pregnancy would have strengthened this analysis; however, half of the mothers included in the study were in their first pregnancy, and all included mothers voluntarily stated their intention to exclusively breastfeed their newborn during hospitalization. Another limitation of our analysis includes breastfeeding education. Last but not least, another limitation is represented by the lack of validation of the LATCH score on the population of mothers in Romania, which is why I think it can represent a new research topic.

## 5. Conclusions

An important result of our study is that among affected mothers, early skin-to-skin contact, increased support and education for mothers during pregnancy and immediately after birth may improve breastfeeding initiation.

The most important finding of this study was that increased salivary cortisol in the last trimester of pregnancy was not associated with delayed initiation/absence of breastfeeding.

The mechanisms explaining the relationship between cortisol levels during pregnancy, maternal psychological distress and suboptimal breastfeeding outcome remain unclear.

These results require future studies on representative subgroups of patients in the first trimester of pregnancy.

## Figures and Tables

**Table 1 medicina-59-00339-t001:** Rating of responses to the HADS Scale, for anxiety or depression.

The Score for Even/Odd Questions Is:				
between 0 and 7 points	→	The subject is in normal limits		
between 8 and 10 points	→	The subject is possibly anxious/depressed		
between 11 and 14 points	→	The subject is moderately anxious/depressive	Total	
between 15 and 21 points	→	The subject is severely anxious/depressed	A	D
The score for all questions is related to emotional stress:				
The higher it is, the higher the distress				

**Table 2 medicina-59-00339-t002:** Explanatory grid used for assessing breastfeeding technique with LATCH score among included patients at 24 and 48 h age after birth.

LATCH Items	0	1	2
Latch	too sleepy or reluctant or no latching achieved	repeated attempts to hold the nipple in the mouth or to stimulate to suck	Grasps breast, Tongue down, Lips flanged Rhythmic sucking
Audible swallowing	ineffective swallowing	a few swallows occurred with stimulation	Audible swallowing occurred spontaneous and intermittent <24 h old spontaneous and frequent >24 h old
Type of nipple	Nipple was inverted	Nipple was flat	Everted nipple was present (after stimulation)
Comfort	Engorged and/or cracked appeared	Filled or reddened breast Small blisters or bruises	Soft and tender breast
Hold	Full assistance was required	Minimal assistance required (staff holds the infant)	No assistance from the staff for mother to hold in good position the infant

**Table 3 medicina-59-00339-t003:** Demographics, obstetrical characteristics, Salivary Cortisol level, HADS * score.

Variables	Mean Anxiety and Depression Scores n = 15, 25% Average Anxiety = 15.25 Average Depression = 15.36	Mild Anxiety and Depression Scores, Respectively, No n = 45, 75% Average Mild Anxiety = 9 Average Depression = 9.1 No Anxiety = 2.41 Without Depression = 2	*p* Value (Group)
Maternal Characteristics			
Maternal age, years, mean (SD)	28.93 ± 6.112 (28.00)	28.36 ± 6.579 (28.00)	0.7656
The environment of origin			
Urban	11	32	RR = 1.087 IC (95%): 0.4010–2.948 *p* = 0.9999
Rural	4	13	
Marital status			
Not married	5	18	RR = 0.8043 IC (95%): 0.3145–2.057 *p* = 0.7639
Married	10	27	
Education			
Secondary education	10	22	RR = 1.750 IC (95%): 0.6792–4.509 *p* = 0.3706
College	5	23	
Social status			
Employed	4	36	RR = 0.1818 IC (95%): 0.6616–0.4997 *p* = 0.0003
Housewife	11	9	
Work in shifts			
Yes	13	19	RR = 5.688 IC (95%): 1.403–23.064 *p* = 0.0031
No	2	26	
States appeared in the 3rd semester of birth			
Pregnancy hypertension			
Yes	2	7	RR = 0.8718 IC (95%): 0.2353–3.230 *p* = 1.000
No	13	38	
Gestational diabetes			
Yes	4	1	RR = 4.000 IC (95%): 2.013–7.949 *p* = 0.0118
No	11	44	
Treatment of chronic diseases			
Yes	4	8	RR = 1.455 IC (95%): 0.5604–3.776 *p* = 0.4725
No	11	37	
Treatment for health states in the 3rd semester of pregnancy			
Yes	4	7	RR = 1.620 IC (95%): 0.6332–4.144 *p* = 0.4423
No	11	38	
Prenatal leave			
Yes	5	10	RR = 1.875 IC (95%): 0.7233–4.861 *p* = 0.2787
No	10	35	
Cortisol level (ng/mL)	54.96 ± 70.73 (26.37)	11.58 ± 5.729 (10.74)	* <0.0001

**Table 4 medicina-59-00339-t004:** Multivariate logistic model for LACH score lower than 8 at 24- and 48-h age.

	OR at 24 h	95.0% Confidence Interval for OR	Value *p*	OR at 48 h	95.0% Confidence Interval for OR	Value *p*
Smokers	−0.105	−1.635–1.424	0.891	0.616	−0.843–2.076	0.401
Parity	−0.213	−1.063–0.638	0.618	−0.226	−1.037–0.586	0.580
Salivary Cortisol level	−0.009	−0.025–0.008	0.296	−0.003	−0.018–0.013	0.710
Cesarean delivery	0.892	−0.509–2.294	0.207	0.438	−0.900–1.775	0.515

**Table 5 medicina-59-00339-t005:** Neonatal characteristics.

Variable Neonatal Characteristics	Average Anxiety and Depression Scores n = 15	Mild Anxiety and Depression Scores, Respectively, no. n = 45	*p* Value
Birth weight, grams, mean ± SD (median)	3139 ± 524.3 (3230)	3242 ± 410.1 (3280)	0.4354
Length, cm, mean ± SD (median)	52.07 ± 2.915 (52.00)	52.49 ± 1.804 (53.00)	* 0.5902
Apgar score at 1 min, mean ± SD (median)	9.333 ± 0.7237 (9.00)	9.044 ± 1.043 (9.00)	* 0.4772
Apgar score at 5 min, mean ± SD (median)	9.800 ± 0.4140 (10.00)	9.511 ± 0.7575 (10.00)	* 0.2141
LATCH score at 24 h mean ± SD (median)	7.000 ± 2.204 (8.00)	7.356 ± 2.506 (8.00)	* 0.4849
LATCH score at 48 h mean ± SD (median)	7.933 ± 2.712 (9.00)	8.600 ± 2.136 (9.00)	* 0.4887

* Test Mann-Whitney.

## Data Availability

The data used and/or analyzed during the current study are available from the corresponding author upon reasonable request. Unfortunately, these data are not publicly available because of privacy or ethical restrictions.

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
