# Peer review of "The Relationship between Psychological Suffering, Value of Maternal Cortisol during Third Trimester of Pregnancy and Breastfeeding Initiation"

_medicina, 2023, doi:10.3390/medicina59020339_

Round 1
Reviewer 1 Report
This study investigated the relationship between salivary cortisol in pregnant women at 0-24 hours before delivery, psychological suffering with the HADS screening test, and successful breastfeeding in the initial postpartum period assessed by the LATCH score at 24 years and 48 hours after delivery.
The justification for this study is not clearly explained in the introductions section, especially the salivary cortisol level during pregnancy. The author did not provide enough background on the change in cortisol levels throughout the pregnancy period in normal and in the stress condition, especially during the third trimester and the parturition period in which the cortisol levels can fluctuate. For example, cortisol levels increase during the onset of labor and decrease rapidly after delivery. Therefore, measurement of salivary cortisol within 24 hours before delivery may not represent the psychological distress that affects successful breastfeeding in the initial postpartum period. It is important to note that cortisol levels during labor and delivery can be influenced by various factors such as stress, anxiety, and pain. Therefore, it should be explained clearly in the introduction section. It needs a clearly define what is the normal limit of salivary cortisol level during each stage of pregnancy and what is the above normal level. In addition, the author did not provide the unit of cortisol in the result (Table 2), thus difficult to estimate the level of salivary cortisol whether it is in the normal range of the parturition or the excessive level from psychological stress. Instead of the measurement of cortisol at a one-time point, measuring cortisol at various time points during the third trimester through parturition might provide more useful information for discussing the relationship between the three factors.
There are lots of typos in the manuscript. Extensive editing of English language and style is required, for examples
Introduction
- The first sentence is incomplete and does not align with the next sentence (page 2, lines 49-50)
- Need more review on the HPA axis activity and cortisol level during pregnancy and parturition period in normal and in stress conditions
Material and methods
- 2.2 page 4, lines 162-165. Please check the sentence, it looks like the response to the reviewer, not the manuscript.
- 2.3 page 5, line 229, 230. Please check that reference 36 should be 37.
Results
- Table 2. Incompletely describe the result in the text.
- Not provide the unit for cortisol level.
Reviewer 2 Report
The authors sought to examine the relationship between psychological state (anxiety/depression), cortisol, and breastfeeding initiation in humans. Overall, the experiment is a good examination of how psychological state could play a role in breastfeeding and the stress status of a mother. However, the authors do not go into detail about the significance or meaning of many of their findings, especially the results that were counter to their hypothesis. This lack of synthesis hinders the power of the paper and the ability of others to understand the scope of the work. Additionally, there were many areas that seemed to be typos, have grammatical errors, or be worded in ways that prevent ease of understanding for the readers. Below are further comments and examples that are aimed to strengthen the paper.
Line 50: This sentence is not complete and is missing a citation for the quote.
Line 56: The and before “decreasing the risk” of should be a comma instead.
Line 61: While the HPA axis has components in the central nervous system, saying the entire axis is a component of the CNS may be an overstatement as the adrenal gland is not typically viewed as part of the CNS.
Line 64: It would strengthen the claim to include why their methodology is debatable
Line 82: I would suggest rewording this sentence as it is currently hard to follow (ie what study is conclusive?).
Line 104: HADS should be spelled out at the first use in the body of the text.
Line 106: You assessed LATCH score at 24 years?
Line 120: Word choice
Line 134-135: An extra ) is present
Line 158: Could the act of hospitalization itself affected the cortisol levels irrespective of anything else? Were you able to take this into consideration?
Line 170: How was HADS implemented?
Line 187: Has this kit been validated for human saliva? If so please include a citation.
Line 200: What was your average coefficient of variation for your duplicates/triplicates of the assay?
Line 236: Did you do comparative analysis to ensure this was the best cutoff (ie trying a cut off of 7 instead as maybe low and high anxiety do not differ, just the presence or absence of anxiety)?
Line 258-259: What do you mean the anxiety scores ranged from (1-20) to 25%? If you mean that 25% of scores ranged from 1-20, I would suggest rewriting this sentence.
Line 254: When describing the statistically significant findings, I would recommend including the direction of effect (ie what is the association between social status and anxiety or depression)? Also, many of these sentences need to be rewritten for clarity and grammar
Line 275: Table 2 is not included in the text of the results as a reference.
Line 287: This should be rewritten so it is not a fragment sentence.
Line 288-291: If something is that statistically insignificant, why are you saying something decreased?
Line 334: Throughout the discussion, I would recommend describing the significance of the findings in addition to what you found (ie why is it important?), especially for the statistically significant findings.
Line 347: If this is important as you say then I would recommend expanding on why it is pertinent to bring up here.
Line 382: Why is this important?
Line 384: You say that these factors were included because of greater evidence of them playing a role in the initiation of breastfeeding, but you don’t talk about the implications of them being nonsignificant here.
Round 2
Reviewer 1 Report
The author respond to all reviewer's comments and the manuscript has been improved.
Author Response
Dear Reviewer,
We appreciate the time and effort that you have dedicated to providing your valuable feedback on my manuscript. We have been able to incorporate changes to reflect most of the suggestions provided.
We look forward to hearing from you in due time regarding our submission and to respond to any further questions and comments you may have.
Yours sincerely,
Anca Rachita
Reviewer 2 Report
R1 Line 200: What was your average coefficient of variation for your duplicates/triplicates of the assay?
A1 The samples being "precious and in small volume" from newborns, for which the recalculation was quite difficult, we did not have enough volume for the duplicate.
R2 I might suggest including that information in the manuscript as it may be important for conclusions drawn.
R1Line 258-259: What do you mean the anxiety scores ranged from (1-20) to 25%? If you mean that 25% of scores ranged from 1-20, I would suggest rewriting this sentence.
A1We made the corrections in the manuscript.
R2 Line 281: rephrasing the sentence to say “ranged from 1-20 for 25% of participants” would make this clearer.
R1 Line 288-291: If something is that statistically insignificant, why are you saying something decreased?
A1 We made the corrections in the manuscript.
R2 If the result is nonsignificant due to the statistical analysis used, you should not say that there is a decrease even if you do not include the statistics for it (was previously p=0.891 and p=0.618 but now the stats are excluded and talked about as if it was significant). Same for the other locations.
R1 Line 288-291: If something is that statistically insignificant, why are you saying something decreased?
A1 We made the corrections in the manuscript.
R2 If the result is nonsignificant due to the statistical analysis used, you should probably not say that there is a decrease even if you do not include the statistics for it (was previously p=0.891 and p=0.618 but now the stats are excluded and talked about as if it was significant). Same for the other locations.
